# Circular Business Models for the Bio-Economy: A Review and New Directions for Future Research

**Wiebke Reim** [1,*], **Vinit Parida** [1,2] **and David R. Sjödin** [1]

1    Entrepreneurship and Innovation, Luleå University of Technology, 971 87 Lulea, Sweden; vinit.parida@ltu.se (V.P.); david.sjodin@ltu.se (D.R.S.)
2    Department of Management, University of Vaasa, FI-65200 Vaasa, Finland
*    Correspondence: wiebke.reim@ltu.se

**Abstract:** Circular and bio-economy represents a political and industrial initiative to ensure that our society can rely on renewable biological sources while achieving economic growth. However, there is a need to critical review how realistic and feasible such initiatives are towards fulfilling the promised benefits of this economy. The literature on bio-economy often discusses the importance of innovative business models and their role in a successful shift to a bio-economy. Still, much of the discussion that is related to circular business models is fragmented and immature. Therefore, the purpose of this study is to conduct a systematic literature review of circular business model activities and the barriers to a bio-economy. Further, this review provides future research directions for a shift to a bio-economy. This study is based on a systematic review of 42 scientific journal articles and book chapters on a forest-based bio-economy. The business model canvas is used to provide a structured aggregation of the existing circular business models activities being used by the forestry sector. In addition, we develop a framework that describes the barriers to bio-economy-based circular business models and suggest new directions for future research. The study highlights the need for alignment among the elements of a business model as a key condition for its successful implementation in a bio-economy.

**Keywords:** bio-economy; circular economy; business models; circular business models; forest; literature review

## 1. Introduction

The shift towards a circular and bio-economy is one of the main focuses of political initiatives that aim to ensure that renewable biological sources are available to society while still achieving economic growth [1]. For example, research shows that the forest sector can contribute significantly to the development of a bio-economy that in turn supports regional development by creating new markets for advanced forest-based products [1–3]. The bio-economy is defined as the production of renewable biological resources and the conversion of these resources and waste streams into value-added products, such as food, bio-based products, and bioenergy, including both traditional and emerging sectors (e.g., agriculture, forestry, food, and pulp and paper production) as well as parts of chemical, biotechnological, and energy industries [4]. The literature on this type of economy devotes much attention to the technological aspects of new bio-based products as well as the political strategies that support a shift toward a sustainable bio-economy [5,6]. However, to succeed with the transition to a bio-economy, there is a need to focus more on the economic feasibility of such initiatives because technological innovations alone are insufficient without a complementary business model innovation [7]. Therefore, a change to a bio-economy requires significant transformation of the innovating company's business model, which requires a redefinition of the value proposition, creation, delivery, and capture [2,8,9]. In contrast to traditional business models, we need circular business

models for the bio-economy that ensure a focus on resources' lifecycles and efficiencies that involve actors beyond the focal company. This study defines a circular business model as one in which a focal company, together with partners, uses innovation to create, capture, and deliver value to improve resource efficiency by extending the lifespan of products and parts that thereby realizes environmental, social, and economic benefits [10]. The literature on the bio-economy often implicitly addresses certain aspects that are connected to circular business models but lacks an important holistic perspective of circular business models, which is needed to achieve a shift towards the bio-economy [11]. Currently, this literature does not fully explore the potential of the circular business model in a bio-economy, and several research gaps exist that show a need to study the bio-economy from the perspective of a business model.

First, much of the discussion related to circular business models for the bio-economy is fragmented across multiple academic disciplines and journals and lacks a thematic organization to drive the research agenda. This study argues that a business model may provide a focal lens to take stock of the literature on the bio-economy and identify key insights and knowledge gaps. A business model perspective is valuable as it inherently focuses on key underlying dimensions of the creation, delivery, and capture of value that are required to take a new bio-based technology from early innovation to full-scale commercialization. Although studies increasingly recognize business-model-related activities to be missing pieces of the puzzle in achieving the benefits of circular business models, none of the prior studies provide an overview of these activities [12–15]. Specifically, the insights from business models related to the bio-economy are unique and currently not fully mapped out.

Second, from a practical perspective, the implementation of a bio-economy is still lagging, and companies are struggling to develop effective business models. Nevertheless, the current literature lacks a relevant research dialogue related to this implementation. Thus, we intend to use the lens of a business model to identify the key actions that companies need to take and the barriers that need to be overcome to enable a successful transition to a bio-based economy. Beyond the barriers that hinder the value proposition, creation, delivery, and capture, a research gap exists on how to the organize the elements of a business model and to align them with each other. This is important because the success or failure of organizations is not determined by the business model elements themselves but rather through their complementarity, interrelations, and alignment [16,17].

This study targets these gaps based on a review of the activities and barriers to circular business models, and we seek to provide direction for future research on circular business models for the bio-economy. Therefore, the purpose of this study is to conduct a systematic literature review of circular business model activities and barriers for the bio-economy and provide future research directions for a shift towards the bio-economy. The study uses a definition of a business model canvas [18] that is based on nine distinctive elements to analyze the literature on forest-based business models in the bio-economy. Furthermore, the sustainable business model framework [12] is used to identify barriers to developing circular business models. The results are integrated in a circular business model framework that focuses on the alignment of business model elements to succeed in a bio-economy.

## 2. Research Method

To advance the understanding of value generation in a bio-economy, the present study consists of a systematic literature review with a specific focus on research related to forest-based business models. For this study, we chose the forest sector as a prominent example of a bio-economy with many ongoing initiatives. Limiting the study to one industry allows an in-depth analysis of the ongoing activities and dynamics related to the bio-economy. According to Cook et al. [19], a systematic review differs from a general review in that it adopts a replicable, scientific, and transparent process. This process leads to developing collective insights based on the theoretical synthesis of existing studies. Therefore, this process limits bias and enhances the legitimacy of the data analysis. These benefits lead to more reliable results that form the basis for drawing conclusions [20].

The literature search used the Scopus database, which is one of the largest multidisciplinary abstract and citation databases of peer-reviewed literature. The database covers research from most major and minor publishers such as Elsevier, Emerald, Springer, and Wiley. Thus, the comprehensiveness of this database allowed us to find studies on bio-economy in the forest sector that have a business or management focus. The study used the three following steps to find relevant articles and to refine the results from the initial search. Figure 1 visualizes the filtering process.

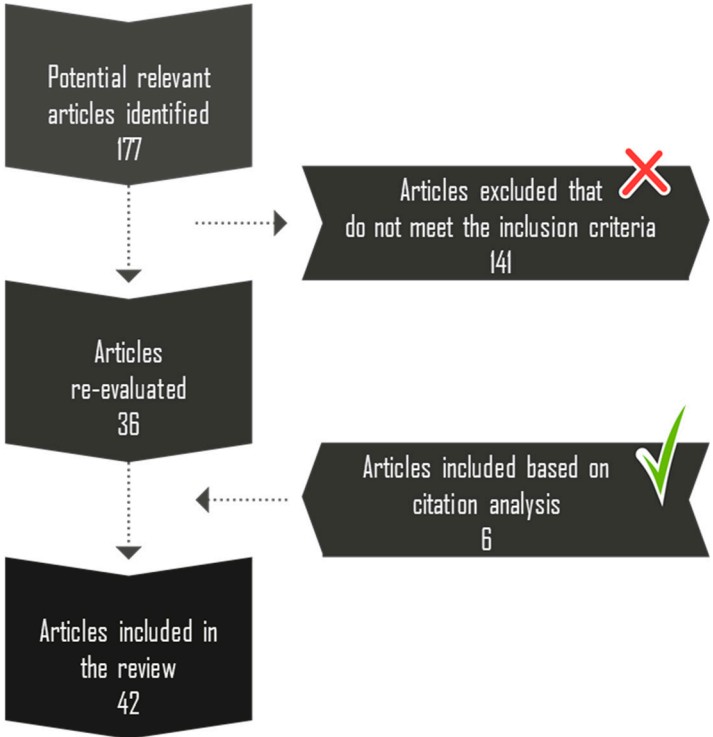

**Figure 1.** Systematic review flow diagram.

*Step 1: Identifying publications and applying practical screening.* The first step started by setting practical screening criteria to ensure that only high-quality publications were included in the review. During the first search, working papers, commentaries, and book reviews were excluded to focus on journal publications, conference articles, as well as book chapters, which are common publication sources in the field of bio-economy. No other quality criteria were used (e.g., journal rankings) for filtering; indeed, publications that cover bio-economy may not always be published in highly ranked journals because this economy is still in its infancy. The search also excluded articles that were not peer-reviewed nor written in English.

The keywords for the literature search were supposed to cover the three areas of interest to the study's purpose. First, bio-economy, which also can be referred to as bio-economy or bio-based economy; either of which should be included in the literature titles, abstracts, or keywords. Second, to identify papers related to the forest sector, the keyword "forest" was used because it would also result in papers that talk about forestry or the forest industry. Third, both the singular and plural forms of "business model" were used as keywords. A search that asked for papers that included all three keywords resulted in only five papers. Therefore, searches combining two of the three keywords were performed. In order to avoid articles that purely focused on technology development in the subject area, our search was limited to studies from business, economics, as well as social and environmental science. This search resulted in 177 articles after eliminating duplicates.

*Step 2: Applying theoretical screening criteria.* Because this study focused on business models in the forest sector, it only included conceptual or empirical studies for further analysis. More specifically, we read all abstracts carefully and retrieved those papers that explicitly or implicitly discussed the

elements of business models. Similarly, even studies that did not explicitly use the term "bio-economy" but discussed new applications or materials as parts of the forest sector were included in the full analysis. After the screening, 36 articles remained.

*Step 3: Final filtering and reference analysis.* In this final stage, all 36 articles that met the inclusion criteria were downloaded and read in detail to analyze their content. Each article's cited references were used as a secondary source of literature analysis. This sourcing identified six additional articles that provided prominent contributions to the understanding of the business models in a forest-based bio-economy. Thus, this systematic literature analysis was based on 42 articles. For analysis of the articles, the study used an open-coding content analysis. When using this analysis, notes and headings were written in the text based on their association with the study's focus. While reviewing the studies, we found that each study could contribute to several different topics. Thereafter, all topics were collected and discussed. This discussion showed that the codes matched very well with the nine elements of the business model canvas [18]. We analyzed each article to determine which of the nine elements it addressed and in what way. By doing this, we were able to discover which elements would require more attention in the future in order to develop a complete business model for a forest-based bio-economy.

## 3. Research Results

The emerging research on a forest-based bio-economy has identified a need for a holistic perspective about business model activities because this would support the commercialization process of bio-based products in the forest sector [2,21]. The definition of a business model focuses on the value proposition, value creation, and delivery as well as value capture [8]. The business model canvas is an important tool to analyze business models in detail, based on nine elements [18]. In the following, we describe the business model activities that focal actors conduct in order to shift to a bio-economy based on the findings from the literature review. An overview of the results is presented in Figure 2.

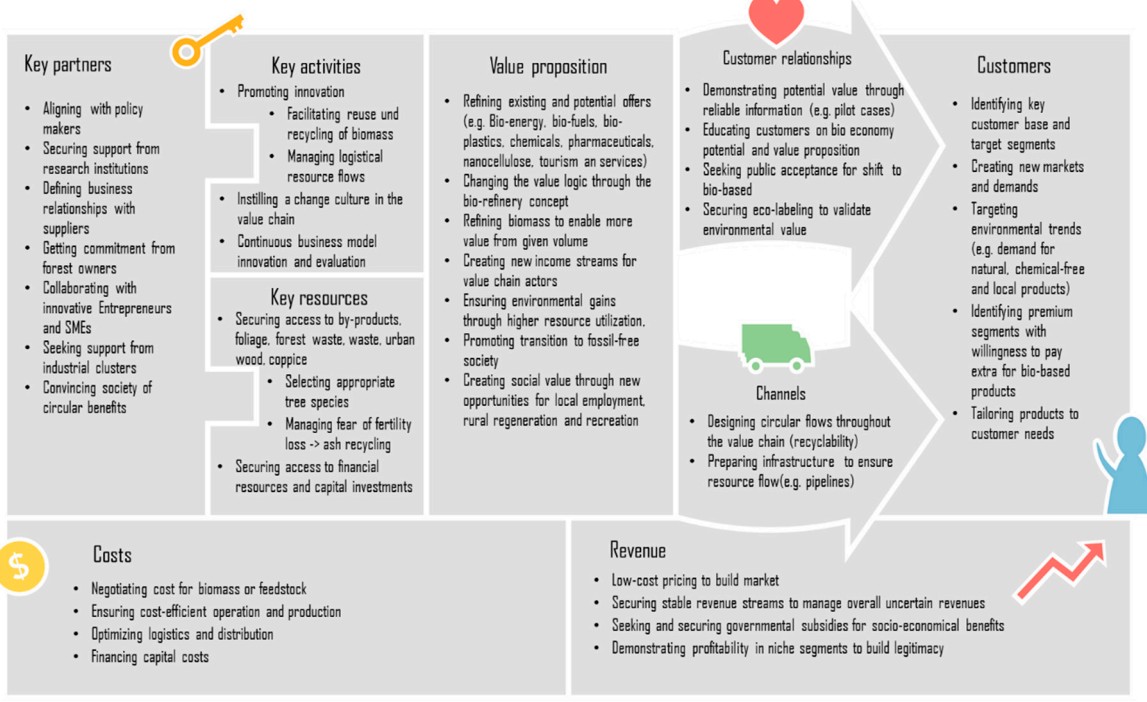

**Figure 2.** Business model canvas for circular business models in a forest-based bio-economy.

### 3.1. Value Proposition

Numerous value proposition activities are described in the literature, mainly when connected to the refining of existing or potential offers related to the bio-economy. This refinement includes offers connected to bio-energy [5,9,22], bio-fuels [6,23,24], bio-plastics [2,24–26], chemicals [22,24,25,27], pharmaceuticals [5], nanocellulose [25,27], and tourism and services [28]. Much of this discussion is related to the concept of bio-refinery, which is frequently studied and should lead to changing value logics [5,25,29–33]. However, an ongoing debate exists on the value generated from bio-refinery outputs, as many argue that the value added from bio-energy and bio-fuels is low [9]. Devappa et al. [5] illustrate this argument by saying that when wood goes up in smoke, it is lost forever. Therefore, a need exists to actively work with the refinement of biomass to enable more value from a given volume.

Considering the economic value generated through a shift to a bio-economy can create new income streams; for example, for pulp and paper companies, a high potential exists to move away from stagnating markets with new business models [27,30,32]. In addition, many articles highlight the environmental value through which the bio-economy can support sustainability [1,6,26,28,34]. Thus, companies should ensure the realization of environmental gains through their business model's design. This design also includes the role of the forest for water protection and bio-diversity [9,35] as well as reduced landfill [36]. Many studies also address the high potential for generating social value that is created through a bio-economy. They frequently mention local employment [6,22–24], rural regeneration [4,24,28,34], recreation [9], and energy security [22,23,34]. Overall, the activities of the focal company need to focus on generating more value from a given volume to ensure economic, environmental, and social value [5,22].

### 3.2. Key Activities

The key activities of companies in a forest-based bio-economy that the literature highlights focus mainly on activities needed to transition to a bio-economy. It frequently mentions the need to promote more innovation [1,2,4,6,24]. An example is the need to further develop and improve the conversion process of various forest-based products and byproducts [22]. To fulfill the goals of a bio-economy, the key activities are to facilitate the reuse and recycling of biomass [37,38]. Another connected and challenging activity is the work on managing logistics, especially when it comes to the reuse of byproducts [31,36]. Furthermore, moving to a bio-economy requires also working on an internal transformation [32], which deals with changing the culture in many traditional forest-based companies [25]. Another key activity that is crucial for success is working continuously with innovating and evaluating the business model [21,28]. In sum, the literature focuses mainly on the activities relevant to the focal company's successful transition to a bio-economy, not its operational activities.

### 3.3. Key Resources

The literature that addresses key resources mainly highlights the importance of securing access to different forest-based inputs for the production processes of the focal companies. The use of byproducts from the traditional production processes in the forest industry has high potential for a forest-based bio-economy [4,5,23,39]. Also, foliage and forest waste that are traditionally unexplored resources come into focus in the move to a bio-economy [5,29,34]. Some of this research focuses on how to utilize waste within a bio-economy [23,29,38] and how urban wood can be used [36]. Furthermore, it examines appropriate tree species [24]; specific tree types (e.g., poplar) are studied [5] as well as the improved utilization of coppice [40,41]. A highlighted downside that companies need to manage is the fear of fertility loss in forests when also harvesting parts that are usually left behind [34]. However, the research proposes ash recycling as an alternative to solve this concern [42]. Even though little is mentioned in regard to intellectual and human resources in the literature, some research highlights that they are important activities to secure access to large financial resources. These resources are needed

for the commercialization of new forest-based products [32] and to manage the high risk that is related to commercialization that can lead to an unwillingness to invest [30,34].

### 3.4. Key Partners

A shift to a bio-economy requires an integrated system of many actors that can all be considered as key partners [23]. Policymakers have placed the goal of a bio-economy high on their agendas and are important key partners who are frequently addressed in the literature [1,4,29,34]. However, regulations currently are unreliable; further, a need exists to integrate public support with the whole value chain to reach ambitious goals [27,29,30]. Other key partners addressed in the literature are research institutions [1,31] and suppliers [22,38], which also include forest owners [34,42]. It is also highlighted that innovative entrepreneurs and SMEs are crucial for the success of a bio-economy [24,30]. Further, a joint effort is needed to facilitate specific clusters that bring actors together [31]. Finally, citizens and overall society should be key partners that are actively involved through responsive governance in the decision-making on how to design a future bio-economy [43].

### 3.5. Customer Relationship

Very little has been written explicitly about how the focal company in the bio-economy should deal with customer relationships. However, reliable information is needed to demonstrate and convince potential customers [6,44], which places high demands on educating and communicating the characteristics and benefits of the bio-economy to the public [26,28]. Public acceptance is crucial to the success of particular products as well as the bio-economy itself [1,38]. One way to facilitate public acceptance and good customer relationships is through securing eco-labeling, as this assures a product's sustainability [1,4,26,35]. However, forest companies are not known for sophisticated marketing and customer interaction, and much room exists for potential improvements [25].

### 3.6. Channels

Channels and related activities represent the business model canvas element that the literature on forest-based bio-economies addresses the least. This lack may be because many studies talk about potential applications for a bio-economy but have not yet reached a stage where the distribution channels receive special attention. Brodin et al. [2] found that the recyclability and reusability of products is important to consider when designing circular flows that include the distribution to and from the customers [28]. Furthermore, Cambero and Sowlati [22] included considerations about the availability and appropriateness of existing or new infrastructure, such as pipelines, to prepare for new products that are connected to the bio-economy.

### 3.7. Customer Segments

To the best of our knowledge, no study has focused specifically on the types of customers in a forest-based bio-economy, and the customer base is still somewhat uncertain [30]. However, there is high potential and need to work with the creation of new markets [4] and to attract new buyers [30,34]. Today's society demands natural, chemical-free, and local products; further, wood-based products fit well with this demand, and these environmental trends should be targeted specifically [25,26]. Customers are willing to pay more for those environmentally friendly products [44], and companies need to identify these premium segments. However, it is important to carefully consider customer needs and market pull [1,21] and to tailor products and services to the potential customers in a bio-economy [25].

### 3.8. Cost Structure

The most frequent explicitly addressed cost in the literature is related to the cost of biomass or feedstock [4,32,34,39]. Therefore, negotiating the cost of biomass is crucial so that a company can

purchase it at a competitive price. However, this negotiation is challenging because competition exists for other uses of biomass [4,22,34]. Furthermore, the company must ensure cost-efficient operation and production in order to be competitive [2,4,32]. It has to optimize logistics and distribution and it has to appropriately finance its capital costs to keep costs at a reasonable level [22,34]. However, Machani et al. [27] stressed the fact that too much focus on traditional cost savings exists instead of focusing on value creation in order to cover the costs.

*3.9. Revenue Streams*

As for revenues, the literature often states that products that are connected to a bio-economy are not yet profitable and that they have to be as cheap as non-bio-based products [2,33,34]. Therefore, low-cost pricing can be used to build markets in the beginning. This strategy could help in securing stable revenues that otherwise would be highly uncertain [30,32]. A bio-economy even offers many possibilities for new revenue streams for existing companies [5]. Seeking and securing governmental subsidies is an important activity to increase the revenue stream [1,25,31,38]. Subsidies, together with taxes, need to account for the socioeconomic benefits that are not included in the market price [34]. However, the companies need to show that the products and services belonging to the forest-based bio-economy are profitable and do not depend only on public support. Some profitable niche segments could build this legitimacy.

## 4. Barriers to Circular Business Model Implementation in a Bio-Economy

Based on the review, the emerging studies have clearly made significant contributions to the circular and bio-economy literature in general and circular business models specifically. These studies present key activities related to implementing circular business models that span across the dimensions of the business model elements. However, to realize the goal of a bio-economy, much remains unclear. Specifically, we need to identify and discuss barriers that hinder the successful implementation of circular business models in a bio-economy. Moreover, building on the discussion of these barriers to circular business models, we are able to pinpoint the key themes that need attention from researchers of business models, the circular economy, and the bio-economy community. Based on the results presented in the previous section and the analysis of existing studies, we developed a prescriptive framework (see Figure 3) that categorizes the barriers based on the four main business model components, that is, value proposition, value creation, value delivery, and value capture. This way of organizing and illustrating barriers was motivated by the sustainable business model canvas from Bocken et al. [12].

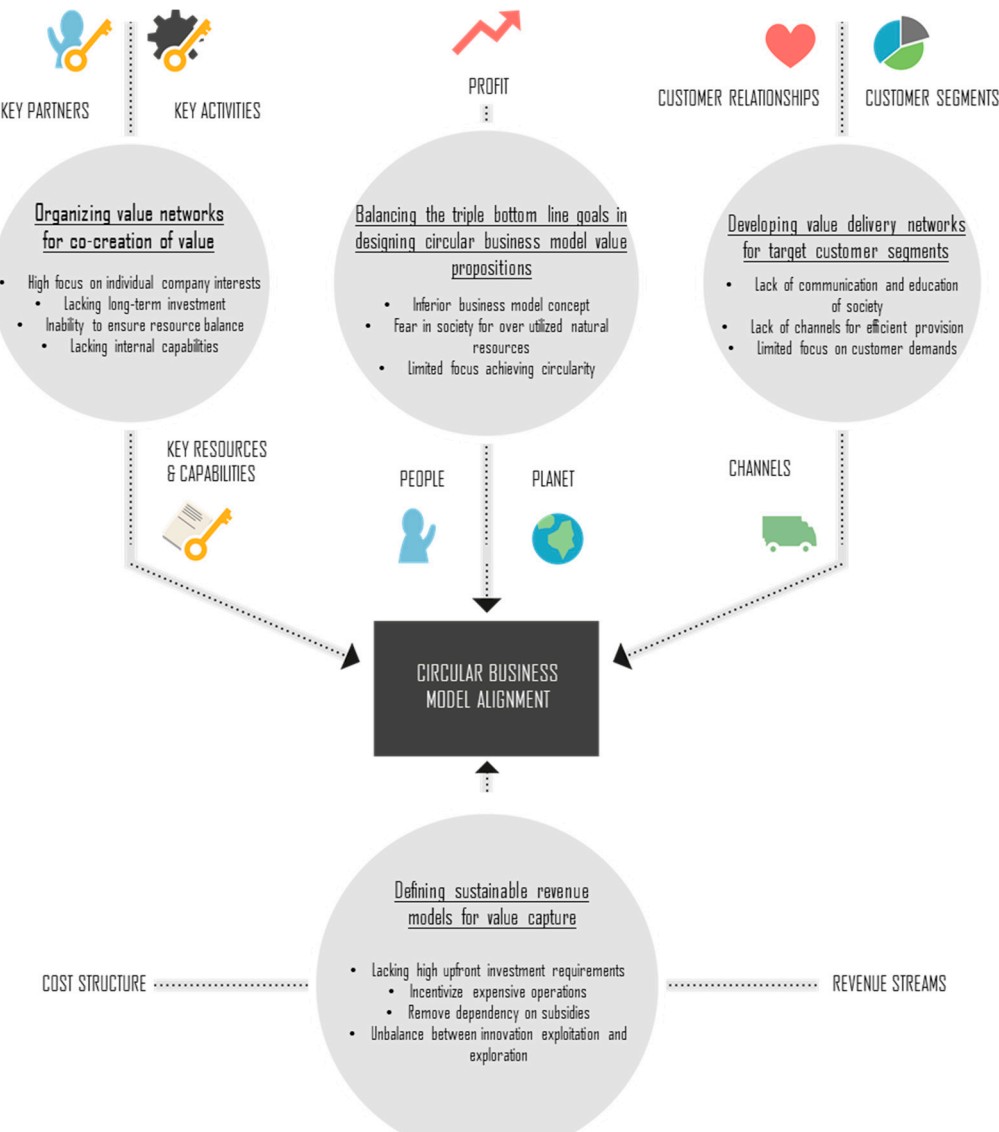

**Figure 3.** Barriers to developing circular business model in a bio-economy.

### 4.1. Balancing the Triple Bottom Line Goals in Designing Circular Business Model Value Propositions

The value proposition describes how a company envisions the value that its product generates. Researchers argue that a bio-economy has a great potential that can lead to economic, social, and environmental benefits. However, they have not designed the commercially offered solutions in a way that ensures the realization of the full potential of circular business models in a bio-economy. A barrier that leads to this unrealized potential is the lack of holistic thinking about the feasibility of the product for the three pillars of people, planet, and profit [8]. Critical to this logic is balancing the three perspectives, since only satisfying one of the principles would likely lead to the other stakeholders being dissatisfied. For example, products that are manufactured from waste or byproducts do not generate value when the logistics are too complex (i.e., too expensive and environmentally consuming) or the customer demand does not exist. Such thinking is crucial because bio-based products compete in the same market with existing traditional non-bio-based products. Another barrier to the realization of social benefits can be that certain stakeholders in a society fear that the bio-economy will overuse natural resources (e.g., forests) to meet the consumption demands for bio-based products that could lead to a tradeoff between sustainability benefits and drawbacks. Moreover, to create the most promising offers, a focus on the circularity in all aspects of the value chain is important to effectively use existing

resources as much as possible. Circularity should not just influence the design of the product itself; it should also include all resources and equipment that a company needs to produce, deliver, and recycle the product. Some research questions that future researchers can focus on related to the value proposition are:

- How to integrate circularity principles for bio-based products into the design of circular business models?
- How to implement the circularity thinking into the design, development, and implementation phases of offer development?
- How to include interests of diverse stakeholders (including government and citizens) from society to be part of circular business models?
- How to develop offers that are commercially attractive without governmental support in order to successfully compete on their own merits with traditional or existing products?

### 4.2. Organizing Value Networks for Cocreation of Value

Value creation categorizes activities, resources, capabilities, and partnerships that are needed to create circular offering. A key barrier to sufficient efficiency is that companies often focus too heavily on internal interests instead of adopting a holistic perspective of the business ecosystem that can lead to the highest possible value creation. The lack of such a perspective within the bio-economy is problematic since the shift to new business models in ecosystems often leads to ambiguity in roles and a lack of clarity on who should take what role in value cocreation [45], which requires focal actors to take a more active role in orchestrating the ecosystem [16]. The critical stakeholders to align are policymakers due to the need for long-term investments that are dependent on stable support from policymakers, as well as diverse investors that believe in the future potential of new radical technologies. Moreover, another barrier to value creation is the need for a new resource balance. In future scenarios of the bio-economy, the inputs to the production processes will change because byproducts that companies have previously used for energy production will be used differently to create value, such as bio-based composite materials. This use means we need to find substitutes for the existing energy needs that these byproducts had met in the past. Thus, this balance in resources is especially crucial to achieve by involving different actors and motivating them to contribute to industry-level transformation. Finally, companies have to develop internal capabilities to manage the transformation to the bio-economy. Companies will need new skills and competences to manage the new technologies and value networks. Some research questions that future researchers can focus on that relate to value creation are:

- How to orchestrate network actors into joining forces for value cocreation? What roles and responsibilities should network or ecosystem actors take for implementation of circular business models?
- How to integrate a resource balance into the design and development of the business model? How to evaluate which alternative uses of resources are most applicable to realize the goals of the bio-economy?
- Which organizational capabilities will be critical in incentivizing network actors to adopt circular business models? How can companies develop missing capabilities?

### 4.3. Developing Value Delivery Networks for Target Customer Segments

Value delivery describes which customer segments a company targets with the product and which channels it uses to reach customers. A main barrier to increase the potential customer base is the lack of clear communication and education. Society is not always aware of the benefits of a bio-economy. Comparisons with traditional offers need to capture the superiority of implementing circular business models. Another barrier is the lack of new and efficient channels for the provision of the circular business model. Channels need to facilitate take-back flows to enable reuse, remanufacturing, and recycling that put a high demand on the development of delivery networks [46]. Traditional companies

do not design sales channels that take such responsibility and often lack the capabilities to develop agreements and manage the logistics chains that fulfil this new role. This lack means that certain actors may need to change their position and roles in the network, and maybe the company needs new actors in its emerging value network. Overall, companies need to focus more on customer demands and their readiness to adopt circular business models. For example, ownerless consumption means that customers only use a product but do not own it. This is a novel and efficient way to increase the use of a product (i.e., shared economy), but many traditional customers are resistant to these business models because of their unfamiliarity with them. Some research questions that future researchers can focus on related to value delivery are:

- How to identify appropriate customer segments for circular business models? What customer characteristics are unique for early acceptance of the goals of a bio-economy?
- How to utilize novel channels for efficient delivery? How can the channel involve new actors (e.g., SMEs) that specialize in higher competence in service delivery?
- How to increase the customers' readiness to adopt a circular business model? What factors can enable behavioral changes in customers?

*4.4. Defining Sustainable Revenue Models for Value Capture*

How much value that a business model captures is determined by the revenue stream, costs, and risk assessment. A major barrier to achieving the capture of sufficient value from high investment comes from the requirements of a bio-economy. The motivation to develop and implement new technologies in the bio-economy requires large upfront investments that its greater uncertainties hinder. The provider of the bio-based offer will thus assume much of the risk that needs to be managed [47], and due to the unclear rules of engagement, opportunistic and potentially costly behavior can occur [48]. Even if the company makes these investments, operations can be costly because of the low volume and the competition for scarce resources. Furthermore, many initiatives in a bio-economy are dependent on subsidies, and economic viability is hard to prove without support from governmental and industrial organizations. Thus, adequate revenue streams are dependent on public financial support that hinders the bio-economy from proving its economic feasibility in the long term. In addition, traditional forest industries usually focus on economies of scale and mainly strive for reducing production costs and increasing volume as much as possible. In the bio-economy, economies of scope are much more important because value should be created by all material flows, which are circular and change characteristics. A balance needs to be found between the focus on the exploitation of existing technologies to create a payback and the need for investment to explore new technologies. Some research questions that future researchers can focus on that are related to value capture are:

- How to design multiactor revenue models for circular business models? How to scale such specialized revenue models from one region to another or from one country to another?
- How to incentivize private and public actors to support the early investment costs? How do they ensure that circular business models survive without favorable support structures?
- How to match value creation with value capturing activities within a traditional industry such as forestry to transform into a significantly different circular business model?

*4.5. Barriers to Circular Business Model Alignment*

The configuration of each element in the business model is very important. However, the success or failure of an organization is determined by the extent of the alignment between each element [8,12]. However, research insights into how to achieve circular business model alignment are largely absent, which opens up new opportunities for multiple research themes for future research.

- How can customer demands, which focus on designing an offer that the customer wants to buy, be aligned with the aim of utilizing the latest and most innovative bio-based technology for a

market that does not exist yet? This dilemma shows a need for alignment between the customer's readiness and acceptance of innovative technology. How can bio-economy industrial dynamics cope with this technological pull and push dilemma?

- How can high and specific customer requirements, which might be difficult for the provider firm to fulfil, such as that the provider takes a certain responsibility or fulfills specific standards (e.g., in terms of environmental impact), be aligned with the new capabilities requirement?

- How can individual and common interests of ecosystem actors be aligned to create a well-functioning business ecosystem to, for example, manage resource flows and competences? How can a company fairly divide the revenue among the ecosystem actors, when certain actors may lose their dominant ecosystem position? How does a company ensure that individual interests are overlooked to achieve the common good through a circular business model in a bio-economy?

- How can the focal actor's interest in exploiting existing technologies be aligned with the need to invest in the exploration of new technologies even though this is related to higher levels of uncertainty, complexity, and ambiguity?

## 5. Conclusions

Driven by the need for a more effective and sustainable way of using our planet's resources, the research on the bio-economy shows that it has a great potential for balancing economic, social, and environmental benefits [22,28]. Still, most products in the bio-economy are not profitable or exist only in a prototype or experimental version because most are still not commercially attractive [2,34]. Therefore, this study proposes that the business model can be a useful tool for examining the status of all aspects relevant for commercialization and sufficient value generation. Through a systematic literature review of studies on a forest-based bio-economy that explicitly or implicitly address circular business models, this study has developed a previously lacking synthesis. This study offers several contributions to research on the implementation of the bio-economy. In particular, our research offers valuable insights into the implementation of circular business models in a bio-economy.

First, this study used the business model canvas with its nine distinctive elements [18] to develop an aggregated view of the key activities that companies need to take to commercialize circular business models in a bio-economy. The high potential of the bio-economy to generate economic, environmental, and social values is well addressed in the literature. However, most of the elements of the business models show a clear need for further investigation because the articles that address them mainly pose questions that need to be answered or problems that need to be solved. For example, policy institutions are the crucial main driver today, but there is a huge need for more research on the total value chain that needs to be considered to succeed with a bio-economy. Comparably, customer relationships are mainly covered by highlighting the need for reliable information and public acceptance. Our study proposed eco-labeling as a way to enhance this process. Channels are the least-addressed element even though new markets and customers are supposed to be attracted by the outcomes of a bio-economy. This connects to the customer segments that emphasize that the customer's needs have to be a focus and that products need to be tailored to bio-economy customers that might be willing to pay a little bit more if the products fulfill their demand for natural, chemical-free, and local products.

Second, the analysis of the elements in the business model showed clear barriers that hinder the success of the commercialization of bio-economy offers. The main barrier to value proposition is the unrealized potential that originates from inappropriate business models that focus too little on circularity aspects. The barriers to value creation center around the need for improved collaboration among partners to balance resources, investments, and capabilities. For value delivery, the barriers are the lack of communication with and education of the potential customers, which is reinforced by the lack of efficient channels and too little focus on analyzing customer demands. Further, the main barriers to value capture are the high expenses for investments and operations as well as the dependence on subsidies because revenue streams are still unsecure.

Finally, a major contribution of this study is the proposal of a research agenda for further understanding the implementation of circular business models in bio-economies. Further, this study contributes to the conceptualization of the most important areas of alignment to develop a successful business model for a bio-economy. This need for alignment also represents the most important areas for future research. Yet, several subissues remain, as illustrated in Section 4. In particular, these issues warrant further examination. More focus is needed on the perspective of a business ecosystem where the generated value needs to be analyzed from a holistic perspective without a focus on every actors' individual value maximization. In addition, more studies are needed on the customers' demands and their readiness to align the developed technologies with the demand in the market. Quantitative studies can make a great contribution here. Innovative revenue models and distribution channels need to be studied to make it possible to cover the high investment costs efficiently.

**Author Contributions:** The individual contributions of the authors were divided as follows: conceptualization: W.R., V.P., and D.R.S.; methodology: W.R.; formal analysis: W.R., V.P., and D.R.S.; writing and original draft preparation: W.R., D.R.S., and V.P.; writing, review, and editing: V.P., D.R.S., and W.R.; supervision: D.R.S.; project administration: V.P.; funding acquisition: V.P., D.R.S., and W.R.

**Funding:** This research received no external funding.

**Acknowledgments:** We are grateful for the support provided by the Luleå University of Technology area of excellence "Effective innovation and organization". The project is entitled Bio-economy-based business models for the forestry industry.

**Conflicts of Interest:** The authors declare no conflict of interest.

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
