# Peer review of "Circular Business Models for the Bio-Economy: A Review and New Directions for Future Research"

_sustainability, doi:10.3390/su11092558_

Round 1
Reviewer 1 Report
Dear Authors,
Thank you for an interesting piece on the circular business models for the bio-based economy.
1. Introduction:
The authors build a gap around lack of business perspective in the current bio-based and advanced forest-based literature. Yet, I'd appreciate if the authors illustrate with a one or two examples what they mean by "too technical approach" and how such approach is insufficient to make a "business case" (position somewhere around line 37-39).
Moreover the author could use the argument from Chesbrough and Rosebloom 2002 who argued that technological innovation alone is insufficient without a complementary business model innovation or change.
Reference
Chesbrough and Rosebloom 2002 ICC The Role of the Business Model in Capturing Value from Innovation -
2. Research Method:
While the authors reported step 1, step 2 and step 3, I'd prefer to see a more detail and transparent figure of the step-by-step process for sample selection. I encourage develop a figure and mirror Okwir et al. 2018: https://onlinelibrary.wiley.com/doi/full/10.1111/ijmr.12184. For instance, in step 1 you claim that only forest sector was considered with a keyword "forest" what is, in my understanding, significant for the reader. A visual illustration would be much appreciated.
Reference:
3. Results:
While Osterwalder's business model canvas is part of the textbooks, it was well-executed and helpful in this particular case.
4. Barriers to Circular Business Model
The figure 2 holds the core value of this paper. Aggregating 9 business blocks around value creation, value capture, value delivery and value proposition came handy.
I propose a few useful references to strengthen some of the aggregate dimensions:
Regarding value creation and capture: A taxonomy of circular economy implementation strategies for manufacturing firms: Analysis of 391 cradle-to-cradle products
Regarding value creation: Managerial practices for designing circular economy business models: The case of an Italian SME in the office supply industry
Regarding value proposition: What do consumers value more in green purchasing? Assessing the sustainability practices from demand side of business
Regarding alignment or the business model components: The Wider Implications of Business-model Research;
5. Conclusion:
While I completely understand the scope and applied research design I'd like to see some reference to the body of literature on technology commercialization as well as technology transfer, going from research bench to commercialization.
Reference to Markman et al 2008 Research and Technology Commercialization
Author Response
COMMENT: Introduction - The authors build a gap around lack of business perspective in the current bio-based and advanced forest-based literature. Yet, I'd appreciate if the authors illustrate with a one or two examples what they mean by "too technical approach" and how such approach is insufficient to make a "business case" (position somewhere around line 37-39).
RESPONSE: Thank you for your comment. We revised the introduction to strengthen the explanation of the concepts we connect to and how they relate to each other. We also explain more explicitly the research gaps that this paper builds on.
COMMENT: Moreover the author could use the argument from Chesbrough and Rosebloom 2002 who argued that technological innovation alone is insufficient without a complementary business model innovation or change.
RESPONSE: This is a very good argument that we now included in the introduction together with a more explicit definition of the used concepts and their connection.
COMMENT: Research Method - While the authors reported step 1, step 2 and step 3, I'd prefer to see a more detail and transparent figure of the step-by-step process for sample selection. I encourage develop a figure and mirror Okwir et al. 2018: https://onlinelibrary.wiley.com/doi/full/10.1111/ijmr.12184. For instance, in step 1 you claim that only forest sector was considered with a keyword "forest" what is, in my understanding, significant for the reader. A visual illustration would be much appreciated.
RESPONSE: Thank you for highlighting the usefulness of a figure. We have added a figure now that shows the step-by-step process and the articles in each step.
COMMENT: Results - While Osterwalder's business model canvas is part of the textbooks, it was well-executed and helpful in this particular case.
RESPONSE: We are happy that you agree that the business model canvas is an appropriate tool in this context.
COMMENT: Barriers to Circular Business Model - The figure 2 holds the core value of this paper. Aggregating 9 business blocks around value creation, value capture, value delivery and value proposition came handy. I propose a few useful references to strengthen some of the aggregate dimensions:
Regarding value creation and capture: A taxonomy of circular economy implementation strategies for manufacturing firms: Analysis of 391 cradle-to-cradle products
Regarding value creation: Managerial practices for designing circular economy business models: The case of an Italian SME in the office supply industry
Regarding value proposition: What do consumers value more in green purchasing? Assessing the sustainability practices from demand side of business
Regarding alignment or the business model components: The Wider Implications of Business-model Research;
RESPONSE: Thank you for these helpful references. We developed the findings based on the recommended literature and some additional references.
COMMENT: Conclusion - While I completely understand the scope and applied research design I'd like to see some reference to the body of literature on technology commercialization as well as technology transfer, going from research bench to commercialization. Reference to Markman et al 2008 Research and Technology Commercialization.
RESPONSE: This is an excellent idea. Especially the literature and technology commercialization is very relevant in this setting and we have integrated that now into the conclusion.
Reviewer 2 Report
In the paper the Authors perform a systematic literature review about circular business models for the bio-economy, taking into consideration that successful shift toward a bio-economy requires economic feasibility and sustainability of particular initiatives.
The Authors apply the business model canvas as a framework for their study.
Although the matter deserves scientific investigation, in my opinion the paper has serious drawbacks.
The title is not adequate to the content of the paper – in the title the Authors don’t mention, that their analysis concern forest sector.
The research procedure is described adequately – particular research stages and their methodologies were presented in detail.
In my opinion, the main weakness of the paper lies in the description of the results. I consider it superficial and too brief – particular elements of the model are referred to in a very general manner. For example,. when discussing value propositions, the Authors mention the example (line142) of the forest for water protection and bio-diversity [7,30] as well as reduced landfill [31]. But what are the particular ideas concerning business value here? The same applies to the rest of the results described by the Authors: they limit their analysis to enumeration of the categories mentioned in the papers they have analysed.The barriers they discuss are not supported by the results.
As a result, in my opinion the value added from the presented literature review is low. For this reason the paper should not be published.
Author Response
COMMENT: In the paper the Authors perform a systematic literature review about circular business models for the bio-economy, taking into consideration that successful shift toward a bio-economy requires economic feasibility and sustainability of particular initiatives. The Authors apply the business model canvas as a framework for their study. Although the matter deserves scientific investigation, in my opinion the paper has serious drawbacks.
RESPONSE: We thank you for your review comments that where helpful in improving the paper. Based on your and the other reviewers comments we have significantly reworked to paper especially when it comes to introduction and presentations of the findings.
COMMENT: The title is not adequate to the content of the paper – in the title the Authors don’t mention, that their analysis concern forest sector.
RESPONSE: You are right that the title and purpose of the paper does not include “forest sector”. The main reason is that we see the forest sector as a representative case of the bio-economy. Choosing a particular case gives us the opportunity to study the concept in more detail while we think that the findings are still relevant for the whole concept of bio-economy. However, we have made this much clearer in both introduction and conclusion.
COMMENT: The research procedure is described adequately – particular research stages and their methodologies were presented in detail.
RESPONSE: We have now even included a figure to visualize the step-by-step process.
COMMENT: In my opinion, the main weakness of the paper lies in the description of the results. I consider it superficial and too brief – particular elements of the model are referred to in a very general manner. For example,. when discussing value propositions, the Authors mention the example (line142) of the forest for water protection and bio-diversity [7,30] as well as reduced landfill [31]. But what are the particular ideas concerning business value here? The same applies to the rest of the results described by the Authors: they limit their analysis to enumeration of the categories mentioned in the papers they have analysed. The barriers they discuss are not supported by the results.
RESPONSE: Thank you very much for this comment, we realized ourselves that the presentation of the results was not sufficient. We reworked that section significantly to make sure that the first part of the results is clearly focusing on the activities that have been highlighted in the literature. The second part on challenges is now more clearly anchored in the literature and the results from the section on business model activities.
COMMENT: As a result, in my opinion the value added from the presented literature review is low. For this reason the paper should not be published.
RESPONSE: Based on the extensive revision of the paper we significantly improved the value of your paper and we would be very happy if you came to the same conclusion.
Reviewer 3 Report
This is a very well written paper and I commend the authors for their review, which is important in the literature. I would like to see some improvements however to this paper. Primarily related to the positioning of it for the circular economy. Now it is a good paper about bio-based economy business models, but not directly applicable/weakly positioned between the two. My comments also are concerned with improving the description of the methods.
General:
1) Typically literature reviews provide a listing of relevant keywords/search strings in the appendix and an analysis complementary to that discussed in the article. Although not always necessary, it would be good to see this.
2) The resuts section begins with a description of the analysis. I would rather see this also in the methodology section to give more input on how you did the search/analysis. Furthermore, is it limited by the fact that you use a predetermined analysis approach (i.e. those specific areas)?
3) The results show a number of references. Are these those found in the literature review? This must be highlighted for the reader to know, otherwise are you only discussing your findings with previous studies?
Specific/Important
It is not that apparent how/why you focus on circular business models and why they are needed for the bio-based economy. (Line 42-46) This needs to be strengthened (see my comment below as well). I find that the text only touches upon circularity/circular economy. It would be beneficial to have a background text outling why/how you include circularity. There is literature available describing why the bio-based cycles are important in the circular economy. This should be strengthened. Even the keywords are not designed to review circularity/circular economy, so this should also be motivated why. Thus, the link to this field is quite weak.
Line 45- "lacking a holistic perspective of all aspects of circular business models."
What does this mean? The paper is about bio-based business models, and why would this be important? I suggest revising or removing.
Author Response
COMMENT: This is a very well written paper and I commend the authors for their review, which is important in the literature. I would like to see some improvements however to this paper. Primarily related to the positioning of it for the circular economy. Now it is a good paper about bio-based economy business models, but not directly applicable/weakly positioned between the two. My comments also are concerned with improving the description of the methods.
RESPONSE: Thank you for your comments that help us improving our paper. Based on your and the other reviewers comments we have significantly reworked to paper especially when it comes to introduction, method and presentations of the findings.
COMMENT: Typically literature reviews provide a listing of relevant keywords/search strings in the appendix and an analysis complementary to that discussed in the article. Although not always necessary, it would be good to see this.
RESPONSE: We have reworked the method section and added a figure to visualize the step-by-step review process. We also describe the used keywords and search strings in a much more detailed level.
COMMENT: The results section begins with a description of the analysis. I would rather see this also in the methodology section to give more input on how you did the search/analysis. Furthermore, is it limited by the fact that you use a predetermined analysis approach (i.e. those specific areas)?
RESPONSE: Thank you for this really good comment. We have restructured the end of the method section and the beginning of the result section. We now also explain that the analysis with help of the business model canvas was not predetermined from the beginning. Rather after the first coding analysis, it became obvious that the business model canvas is a very appropriate tool to structure and visualize the findings.
COMMENT: The results show a number of references. Are these those found in the literature review? This must be highlighted for the reader to know, otherwise are you only discussing your findings with previous studies?
RESPONSE: Thank you for pointing this out. We have now clarified that the results and the references in the result sections come from the literature review. We have also marked all references that are part of the literature review in the reference list with an *.
COMMENT: It is not that apparent how/why you focus on circular business models and why they are needed for the bio-based economy. (Line 42-46) This needs to be strengthened (see my comment below as well). I find that the text only touches upon circularity/circular economy. It would be beneficial to have a background text outling why/how you include circularity. There is literature available describing why the bio-based cycles are important in the circular economy. This should be strengthened. Even the keywords are not designed to review circularity/circular economy, so this should also be motivated why. Thus, the link to this field is quite weak.
RESPONSE: We have significantly reworked the introduction to especially clarify the relationship between circular economy, bio-economy and circular business models. This argumentation is built based on definitions and literature.
COMMENT: Line 45- "lacking a holistic perspective of all aspects of circular business models." What does this mean? The paper is about bio-based business models, and why would this be important? I suggest revising or removing.
RESPONSE: Thank you for pointing out this vagueness. We reformulated the sentence into: ” Current literature on bio-economy often implicitly addresses certain aspects connected to circular business models but is lacking the important holistic perspective of circular business models, which is needed to achieve a shift towards the bio-economy”
Reviewer 4 Report
Thank you for allowing me to review your manuscript titled “Circular Business Models for the Bio-Based Economy: A Systematic Literature Review.” I would like to applaud the authors for their efforts as the paper is well written and has high relevancy in today’s economies. With that said, please allow me to recommend ways to improve the paper.
The abstract can be shortened extensively. Try to be more parsimonious with how much text is in the abstract. Some of what you say is an exact repeat of text in the introduction. I believe you can get the same points across with fewer words.
As a whole, I think the introduction needs to be more clear and succinct. The authors briefly mention important aspects of the topic and how this paper is relevant, but more depth would elevate the paper.
For example, it would help the reader if you explained what a bio-economy is in the introduction and why it is important, more so than is explained now (e.g. supports regional development).
In the introduction you explain that focus needs to be on designing circular business models rather than traditional business models. This should be explained more clearly and as to why.
The framework of the intro is fine, but more depth or expansion would be helpful especially since you move directly into the research method.
Did you follow a similar method to finding studies based on previous research? Please explain.
Please revise paragraph 2 of Step 1. The middle to end of the text of the paragraph becomes confusing.
Page 9 Conclusion paragraph 1 – can you explain the following sentence further: “Still most bio-economy offers are not profitable or exist only in a prototype version because commercialization is not feasible.” That is somewhat confusing. Besides that portion of the back end of the paper, I do not have any issues. The authors explained fully the context and the quality was high.
Overall, I believe the paper is of high quality, but the front end can be improved. The introduction needs to be extrapolated and have more depth attached to it. The bare bones framework is there, but more explanation is needed.
Minor note: there were a few grammatical errors throughout.
Thank you again for allowing me to review your paper. Good luck on developing this paper and best of luck moving forward with your research.
Author Response
COMMENT: Thank you for allowing me to review your manuscript titled “Circular Business Models for the Bio-Based Economy: A Systematic Literature Review.” I would like to applaud the authors for their efforts as the paper is well written and has high relevancy in today’s economies. With that said, please allow me to recommend ways to improve the paper.
RESPONSE: Thank you for your comments that help us at lot with improving our paper. Based on your and the other reviewers comments we have significantly reworked to paper especially when it comes to introduction, method and presentations of the findings. In the following, we outline the changes we made in detail.
COMMENT: The abstract can be shortened extensively. Try to be more parsimonious with how much text is in the abstract. Some of what you say is an exact repeat of text in the introduction. I believe you can get the same points across with fewer words.
RESPONSE: Thank you for pointing this out. We have shorten the abstract significantly to only include the most relevant information.
COMMENT: As a whole, I think the introduction needs to be more clear and succinct. The authors briefly mention important aspects of the topic and how this paper is relevant, but more depth would elevate the paper. For example, it would help the reader if you explained what a bio-economy is in the introduction and why it is important, more so than is explained now (e.g. supports regional development).
RESPONSE: We have significantly reworked the introduction to especially clarify the relationship between circular economy, bio-economy and circular business models. This argumentation is built based on definitions and literature. We also present the research gaps more explicitly.
COMMENT: In the introduction you explain that focus needs to be on designing circular business models rather than traditional business models. This should be explained more clearly and as to why.
RESPONSE: After revising the introduction this connection is now much better explained.
COMMENT: The framework of the intro is fine, but more depth or expansion would be helpful especially since you move directly into the research method.
RESPONSE: As we said before, we have significantly reworked the introduction to especially clarify the relationship between circular economy, bio-economy and circular business models. This argumentation is built based on definitions and literature. We also present the research gaps more explicitly. This is done especially with having in mind that the paper has no theoretical background section.
COMMENT: Did you follow a similar method to finding studies based on previous research? Please explain.
RESPONSE: In the method section we now clearly state that we follow an existing method and that we also used the literature we found to identify even more papers.
COMMENT: Please revise paragraph 2 of Step 1. The middle to end of the text of the paragraph becomes confusing.
RESPONSE: Thank you for pointing this out. The paragraph is now revised.
COMMENT: Page 9 Conclusion paragraph 1 – can you explain the following sentence further: “Still most bio-economy offers are not profitable or exist only in a prototype version because commercialization is not feasible.” That is somewhat confusing. Besides that portion of the back end of the paper, I do not have any issues. The authors explained fully the context and the quality was high.
RESPONSE: The sentence is now revised.
COMMENT: Overall, I believe the paper is of high quality, but the front end can be improved. The introduction needs to be extrapolated and have more depth attached to it. The bare bones framework is there, but more explanation is needed. Thank you again for allowing me to review your paper. Good luck on developing this paper and best of luck moving forward with your research.
RESPONSE: We thank you again for our comments that help us a lot in improving the paper.
Reviewer 5 Report
Dear Authors,
The idea that informed the article is good and fits within the aims and scope of Sustainability. However, the practical implementation of the study, research outcomes and its implications need to be seriously thought through by Authors.
In the introduction, the motivations behind the selection of the forestry sector and of the systematic literature review method are not robust enough. Research gaps and the contribution of your study are not clear.
A theoretical framework is missing; my suggestion is starting from a literature review about existing systematic reviews in the bioeconomy and/or forestry sectors, for example by searching for “systematic review bioeconomy” on Google Scholar.
The explained method does not fit the definition of systematic review. For example, no clear question is presented, just one database is used as data source, the search string(s) is not shown, the PRISMA diagram is not shown (e.g. as required by Sustainability for systematic reviews), etc.
The list of the selected articles for review is missing.
The open-coding procedure is not presented; a code list with the relative definitions is also needed. Inductive coding should be motivated, especially when compared to deductive coding. Are you sure that you could not adopt a deductive coding approach? How did you use the codes?
Raw coding results should be presented.
Results are very general and miss any comparison with existing studies.
Business models are never described and forestry-specificity should be highlighted.
Author Response
COMMENT: The idea that informed the article is good and fits within the aims and scope of Sustainability. However, the practical implementation of the study, research outcomes and its implications need to be seriously thought through by Authors.
RESPONSE: Thank you for your comments that help us at lot with improving our paper. Based on your and the other reviewers comments we have significantly reworked to paper especially when it comes to introduction, method and presentations of the findings. In the following, we outline the changes we made in detail.
COMMENT: In the introduction, the motivations behind the selection of the forestry sector and of the systematic literature review method are not robust enough. Research gaps and the contribution of your study are not clear.
RESPONSE: We have significantly reworked the introduction to especially clarify the relationship between circular economy, bio-economy, circular business models and the relevance of forest sector for it. This argumentation is built based on definitions and literature. We also present the research gaps more explicitly.
COMMENT: A theoretical framework is missing; my suggestion is starting from a literature review about existing systematic reviews in the bioeconomy and/or forestry sectors, for example by searching for “systematic review bioeconomy” on Google Scholar.
RESPONSE: Thank you for your comment. We recognized as well that the theoretical foundation was not strong enough. We have now included a much stronger theoretical foundation in the introduction to show the current stare of research in field and how the studied fields are linked to each other.
COMMENT: The explained method does not fit the definition of systematic review. For example, no clear question is presented, just one database is used as data source, the search string(s) is not shown, the PRISMA diagram is not shown (e.g. as required by Sustainability for systematic reviews), etc.
RESPONSE: Thank you for pointing this out. We explain and visualize now in more detail how we did the literature review to fulfill the criteria.
COMMENT: The list of the selected articles for review is missing.
RESPONSE: We also realized that we did not present the articles that are included in the review. We now mark all articles that are cited and part of the review with an * in the reference list.
COMMENT: The open-coding procedure is not presented; a code list with the relative definitions is also needed. Inductive coding should be motivated, especially when compared to deductive coding. Are you sure that you could not adopt a deductive coding approach? How did you use the codes? Raw coding results should be presented.
RESPONSE: Thank you for this really good comment. We have restructured the end of the method section and the beginning of the result section to clarify our data analysis. We now also explain that the analysis with help of the business model canvas was not predetermined from the beginning. Rather after the first coding analysis, it became obvious that the business model canvas is a very appropriate tool to structure and visualize the findings. The raw codes are very close to the content that is presented within the business model canvas blocks.
COMMENT: Results are very general and miss any comparison with existing studies.
RESPONSE: Thank you for your comment. We have now both in the conclusion and barriers framework increased the comparison with existing literature. We have also reworked the result section with a more detailed focus on the business model activities that are presented in the literature and how this connects to the identified barriers.
COMMENT: Business models are never described and forestry-specificity should be highlighted.
RESPONSE: In line with you comment we realized that the introduction needs a better argumentation, which defines different concepts and links them to each other. This is now done.
Round 2
Reviewer 2 Report
I believe the corrected version of the paper warrants publication in 'Sustainability': The Authors clarified the scope of their study (which was one of my objections) and described methodology and results in more detail. The paper needs spell check (I understand this is because of the short time for revision). My only suggestion for the Authors is to add a paragraph considering policy implications. In fact, the policy (or lack of it) is at the roots of the most barriers they name and the policy should be changed to create incentives for businesses related to circular bio-economy.
Author Response
Thank you for your comments and support. We have now also highlighted the implications for policy development.
Reviewer 3 Report
As in the first revision, I think the paper was very well written and a timely subject. The authors have significantly improved the article based on my own, and the comments of the other reviewers. It now provides a much more coherent and transparent message. I am happy to accept the article in this case.
Author Response
Thank you for your comments and support.